# Predicting What Matters: Robust Generalist Robot Policy Learning via Future Semantic Mask

**Yunfan Lou** [* 1 2] **Xiaowei Chi** [* 3] **Xiaojie Zhang** [* † 2 4] **Zezhong Qian** [4] **Chengxuan Li** [4] **Rongyu Zhang** [† 5 6]
**Yaoxu Lyu** [2 4] **Guoyu Song** [5] **Chuyao Fu** [2] **Haoxuan Xu** [3] **Pengwei Wang** [2] **Shanghang Zhang** [✉ 2 4]

## Abstract

World models derived from large-scale video generative pre-training have emerged as a promising paradigm for generalist robot policy learning. However, standard approaches often focus on high-fidelity RGB video prediction, this can result in overfitting to irrelevant factors, such as dynamic backgrounds and illumination changes. These distractions reduce the model's ability to generalize, ultimately leading to unreliable and fragile control policies. To address this, we introduce the **Mask World Model (MWM)**, which leverages video diffusion architectures to predict the evolution of semantic masks instead of pixels. This shift imposes a geometric information bottleneck, forcing the model to capture essential physical dynamics and contact relations while filtering out visual noise. We seamlessly integrate this mask dynamics backbone with a diffusion-based policy head to enable robust end-to-end control. Extensive evaluations demonstrate the superiority of MWM on the LIBERO and RLBench simulation benchmarks, significantly outperforming the state-of-the-art RGB-based world models. Furthermore, real-world experiments and robustness evaluation (via random token pruning) reveal that MWM exhibits superior generalization capabilities and robust resilience to texture information loss.

[*]Equal contribution  [†] Project leaders  [✉] Corresponding author
[1]National University of Singapore, Singapore [2]Beijing Academy of Artificial Intelligence, Beijing, China [3]The Hong Kong University of Science and Technology, Hong Kong, China [4]State Key Laboratory of Multimedia Information Processing, School of Computer Science, Peking University [5]Peking University, Beijing, China [6]Nanjing University, Nanjing, China. Correspondence to: Shanghang Zhang <shanghang@pku.edu.cn>.

*Proceedings of the 43ʳᵈ International Conference on Machine Learning*, Seoul, South Korea. PMLR 306, 2026. Copyright 2026 by the author(s).

## 1. Introduction

Learning generalist robot manipulation policies that remain reliable under real-world visual variability remains a central challenge. A promising direction is to learn a predictive world model and use its internal features to guide action generation (Hu et al., 2025; Assran et al., 2025; Ha & Schmidhuber, 2018; Hafner et al., 2020; Chi et al., 2025c), offering long-horizon foresight and improved data efficiency (Pai et al., 2025). However, most video world models are optimized to predict RGB pixels, and this photometric objective is often misaligned with control.

RGB frames contain substantial nuisance variation, including texture, lighting, reflections, and dynamic backgrounds, which is weakly related to action selection (Wang et al., 2023). Pixel prediction compels a model to allocate capacity to these factors and to entangle appearance with dynamics, treating changes in illumination or background as comparable to contact-relevant motion (Grooten et al., 2023; Wang et al., 2025). In closed-loop execution, this misallocation becomes more damaging: small appearance-driven errors can compound over time, causing predictive drift and brittle policies under modest distribution shifts (Liu et al., 2025).

We argue that robust control benefits from predicting decision-relevant dynamics rather than photometric realism (Chi et al., 2024; Ze et al., 2024). To this end, we introduce the MWM (Figure 1), which shifts the prediction space from future RGB frames to future semantic masks. Semantic masks impose a geometric bottleneck that preserves object identity, spatial layout, and interaction-relevant structure while discarding redundant appearance (Seo et al., 2023; Berg et al., 2025). Crucially, MWM does not require an external segmentation model at inference: semantic labels are used only offline during training, while deployment uses only raw multi-view RGB (Zhu et al., 2025).

The training pipeline of the proposed MWM adopts a two-stage strategy, as illustrated in Figure 1. First, we learn a mask-centric predictive model by forecasting future semantic masks with a conditional diffusion objective, so the predictor models the evolution of semantic structure instead of reconstructing pixels. Second, we train a diffusion policy

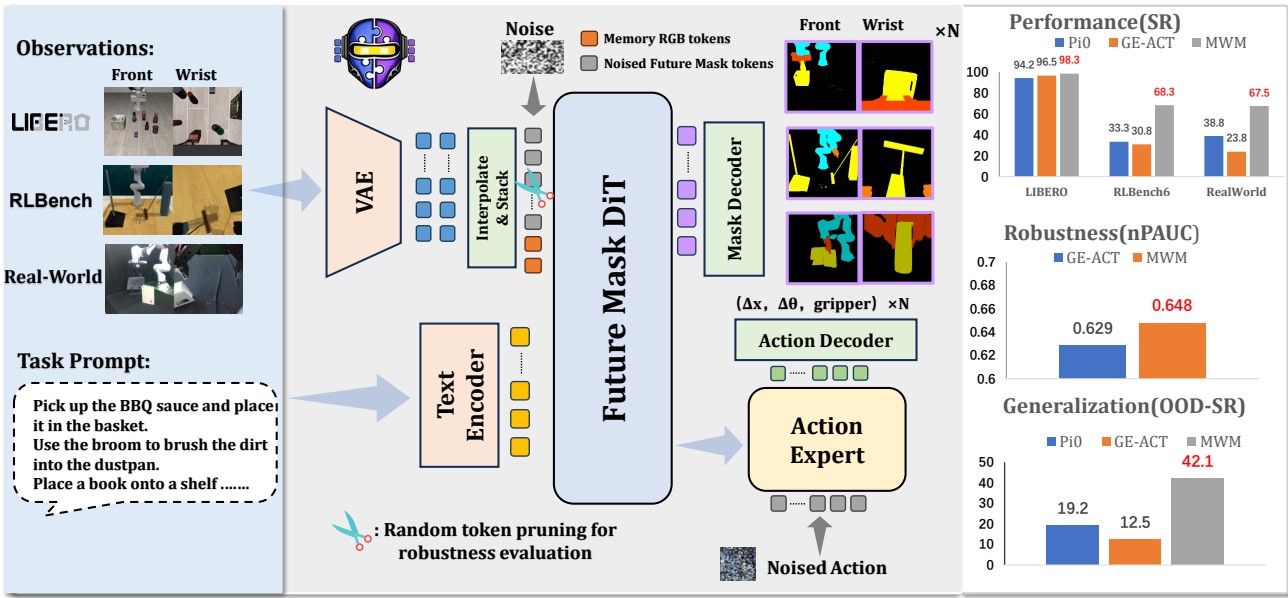

*Figure 1.* **MWM overview.** MWM learns a mask-centric predictive world model from semantic supervision during training, but runs purely on raw multi-view RGB at test time. Training proceeds in two stages: we first learn to forecast future semantic masks via conditional diffusion, then train a diffusion policy that conditions on mask-centric predictive features for action generation. This semantic bottleneck prioritizes decision-relevant geometry and interaction dynamics over photometric realism.

that conditions on intermediate features produced by the mask-centric model to generate actions. This coupling is essential: *the policy is explicitly optimized to extract control-utility from mask-centric predictions, rather than treating masks as an auxiliary visualization.* Empirically, we find that the semantic bottleneck improves both representation quality and control. This is because mask-centric predictive features focus on capturing motion and contact geometry while filtering out irrelevant photometric variations, providing stronger and more reliable action guidance compared to RGB-centric features (He et al., 2021).

Extensive experiments demonstrate that MWM yields consistent gains across multiple domains with 98.3% average success rate on LIBERO, 68.3% on RLBench, and 67.5% average success on a real Franka robot across four tasks, substantially outperforming strong RGB-centric baselines. In addition, we observe improved robustness and generalization under controlled real-robot appearance shifts (background, lighting, object color), as well as under compute/observability stress tests such as random visual token pruning. Our main contributions can be concluded as:

1. We propose **Mask World Model**, a world model that forecasts future semantic masks internally and runs purely on raw RGB at inference, using semantic supervision only during training to eliminate reliance on external segmenters at deployment.

2. We design a **mask-guided diffusion policy** that conditions action generation on mask-centric predictive

features, and show that these representations provide stronger control utility than RGB-centric prediction by emphasizing dynamics-relevant structure.

3. We investigate several approaches to extracting and utilizing mask information and observe that mask-centric designs consistently outperform future RGB prediction, which highlights that the performance gains primarily arise from the shift in representation and objectives, rather than reliance on a specific architectural design.

## 2. Related Work

### 2.1. Video world models for robot policy learning

World models have long been used to learn compact predictive dynamics for control, often in latent spaces to avoid direct pixel prediction (e.g., Dreamer-style agents) (Hafner et al., 2024; 2025). More recently, diffusion/transformer video generators have been repurposed as predictive backbones for robotics and physical simulation (NVIDIA et al., 2025; Brooks et al., 2024), and some systems couple multi-view video diffusion with action heads for manipulation (Liao et al., 2025; Chi et al., 2025b;a; Li et al., 2025c; Mi et al., 2026; Jia et al., 2026). Despite strong results, these methods typically optimize prediction in an appearance-centric space (RGB reconstruction or closely related latents), which encourages modeling nuisance variation (texture, lighting, background) and can entangle appearance with interaction dynamics. Concurrent diffusion world-model variants improve scalability or handle multi-modality

(Huang et al., 2025b; Shang et al., 2025; Bu et al., 2025; Qian et al., 2025; Li et al., 2025b), but their predictive targets remain primarily photometric. In contrast, MWM predicts future semantic mask dynamics, using semantic supervision only during training and requiring no external segmenter at test time.

## 2.2. Vision-Language-Action models

Generalist Vision-Language-Action (VLA) policies leverage large pretrained vision-language representations to map instructions and observations to actions (Brohan et al., 2023; Team et al., 2024; Zhang et al., 2025b;a). When tasks require precise spatial relations and contact-sensitive control, policies can benefit from representations that more explicitly expose object state and interaction geometry and are less coupled to photometric variation. A growing line of work injects semantics to reduce perceptual redundancy or improve grounding, e.g., instruction-aligned token pruning (Li et al., 2025a), auxiliary reconstruction to emphasize task-relevant regions (Song et al., 2025), or externally produced grounding masks as intermediate cues (Huang et al., 2025a). MWM differs in both mechanism and deployment: rather than using semantics as an input cue on the current observation, it provides semantic lookahead by learning predictive mask dynamics and trains a policy head to consume mask-centric predictive features for action generation, while retaining a pure-RGB interface at test time.

## 2.3. Structured representations under masking

Object-centric representations and scene factorization aim to discover structured entities and dynamics (Locatello et al., 2020; Burgess et al., 2019; Greff et al., 2020; Engelcke et al., 2022), while masked modeling and token dropping are widely used to learn robust and efficient visual representations (He et al., 2021; Tong et al., 2022). These lines support the broader intuition that compact, structure-biased representations can improve stability over raw pixels. MWM instantiates this idea for manipulation by making semantic structure the predictive space of a diffusion world model and coupling it to a diffusion policy head. We further adopt random visual token pruning as an evaluation stress test (not a modeling contribution) to probe control robustness under severe partial observability.

## 3. Method

We present the **MWM**, a mask-centric world-model for language-conditioned manipulation. As shown in Figure 2, MWM consists of a mask-dynamics backbone and a diffusion policy head. Training follows two stages: (i) pretrain the backbone to forecast future semantic mask latents using offline semantic supervision, and (ii) train the action diffusion head for control, where gradients from $\mathcal{L}_{\text{act}}$ also

update the backbone to make predictive features maximally action-relevant. At test time, MWM takes only multi-view RGB as input without external segmentation model.

## 3.1. Motivation: Geometric Information Bottleneck

Robust manipulation relies on object identity, spatial layout, and contact-relevant dynamics, while RGB factors such as texture, illumination, and background often act as nuisance variables. By predicting future semantic masks, MWM introduces a geometric bottleneck that preserves decision-critical structure while filtering out irrelevant photometric variation.

## 3.2. Problem Setup

At time $t$, the agent receives multi-view RGB observations $\mathbf{o}_t = \{\mathbf{o}_t^{(v)}\}_{v=1}^V$ and a language instruction $\mathbf{p}$, and outputs a continuous action $\mathbf{a}_t$ (end-effector motion and gripper command). We learn a closed-loop policy robust to appearance variations (e.g., background, lighting, and object color), where such photometric changes should not dominate learned dynamics.

## 3.3. Mask Dynamics Backbone

**Offline semantic supervision; RGB-only at deployment.** During training, each frame is paired with a semantic mask $\mathbf{m}_t \in \{0, 1\}^{H \times W \times C}$ for task-relevant entities (robot arm/gripper and manipulated objects). These masks are used *only* to supervise the backbone predictor; at deployment, the model takes only raw RGB as input.

**Discrete-to-continuous mask encoding with a shared VAE.** Semantic masks are discrete, while video diffusion backbones typically operate on continuous latents. MWM reuses the *same pretrained video VAE* (Kingma & Welling, 2022) for RGB and mask targets by rendering masks into an RGB-compatible image $\tilde{\mathbf{m}}_t \in [0, 1]^{H \times W \times 3}$ using a fixed color palette (each semantic region mapped to a distinct color). We then encode both RGB frames and rendered masks with the same encoder $\mathcal{E}$:

$$\mathbf{z}_t^o = \mathcal{E}(\mathbf{o}_t), \qquad \mathbf{z}_t^m = \mathcal{E}(\tilde{\mathbf{m}}_t). \qquad (1)$$

This design avoids modifying the pretrained VAE and yields a consistent latent interface for context (RGB) and targets (masks).

**Latent normalization, Interpolate and Stack.** Let $\mathbf{z} \in \mathbb{R}^{T \times H' \times W' \times D}$ denote VAE latents. We apply channel-wise normalization using VAE statistics:

$$\bar{\mathbf{z}} = (\mathbf{z} - \boldsymbol{\mu}_{\text{VAE}}) \oslash \boldsymbol{\sigma}_{\text{VAE}}. \qquad (2)$$

To obtain a fixed-length token sequence, we temporally resample (interpolate) latents to a canonical latent frame rate

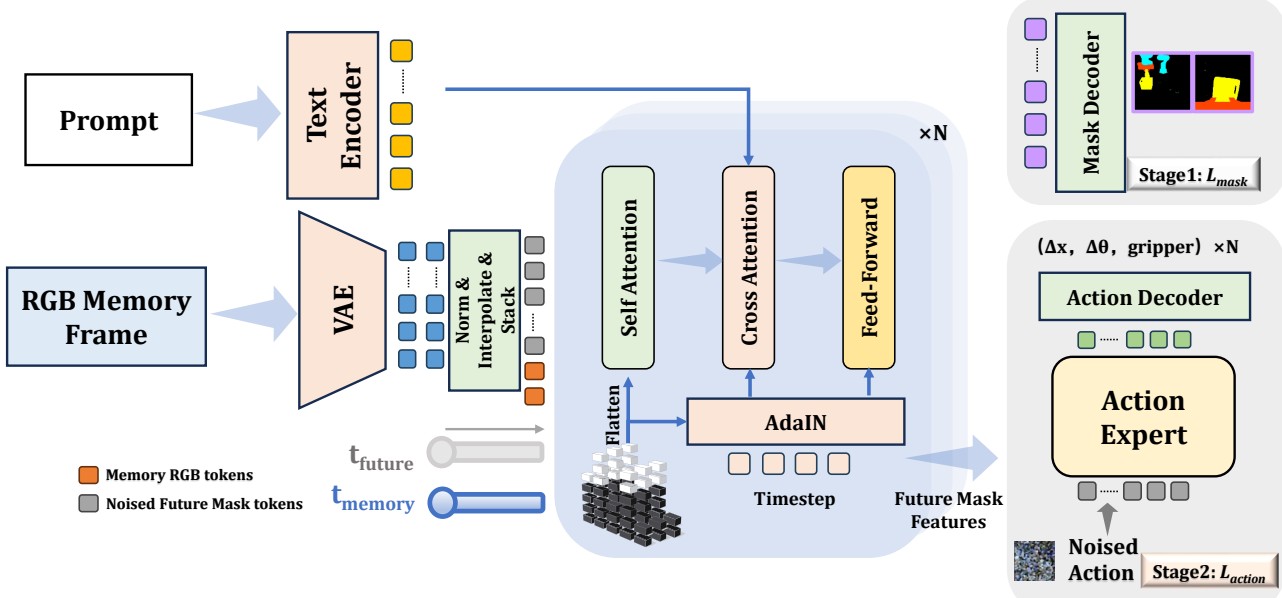

*Figure 2.* **Mask World Model (MWM) architecture.** Given multi-view RGB memory frames and a language prompt, MWM encodes observations with a shared video VAE, then applies *Normalize & Interpolate & Stack* to form a fixed-length latent token sequence. A DiT-style backbone with AdaIN timestep conditioning and text cross-attention processes these tokens for $N{=}28$ transformer blocks. In Stage 1, a mask decoder supervises future mask latent prediction with $\mathcal{L}_{\text{mask}}$. In Stage 2, an action diffusion head (action expert + action decoder) attends to the backbone predictive features via cross-attention and is trained with $\mathcal{L}_{\text{act}}$ only.

and stack multi-view latents into a unified sequence before flattening into tokens. Concretely, for view $v$ we resample $\bar{\mathbf{z}}^{(v)}$ along time to $\hat{\mathbf{z}}^{(v)} \in \mathbb{R}^{\hat{T} \times H' \times W' \times D}$, then stack over $v$ and flatten spatially into tokens for a transformer backbone. This step replaces the conventional pixel-space 3D patchification, enabling more stable training even with varying temporal subsampling rates and compression artifacts brought by the VAE module.

**Conditional flow matching with fixed memory slots.** Given a memory window of RGB latents $\hat{\mathbf{z}}^o_{t-n+1:t}$ and target future mask latents $\hat{\mathbf{z}}^m_{t+1:t+\tau}$, the backbone learns to generate future mask latents conditioned on the memory and language. We use flow matching(Lipman et al., 2023) with a linear interpolation path between clean target latents and noise. Let $\mathbf{z}_0$ be clean future mask latents and $\mathbf{z}_1 \sim \mathcal{N}(\mathbf{0}, \mathbf{I})$. For $s \in [0,1]$,

$$\mathbf{z}_s = (1-s)\mathbf{z}_0 + s\mathbf{z}_1. \tag{3}$$

Crucially, conditioning enters through the *velocity field*, not the path: the transformer predicts $\mathbf{v}_\theta(\mathbf{z}_s, s, \mathbf{c}_t)$ where $\mathbf{c}_t$ encodes the RGB memory and text instruction and is injected via cross-attention. The loss is

$$\mathcal{L}_{\text{mask}} = \mathbb{E}\Big[ w(s) \big\| \mathbf{v}_\theta(\mathbf{z}_s, s, \mathbf{c}_t) - (\mathbf{z}_1 - \mathbf{z}_0) \big\|_2^2 \Big]. \tag{4}$$

**Conditioning mask (memory clean; future denoised).** We represent the latent sequence as $[\hat{\mathbf{z}}^o_{t-n+1:t}, \hat{\mathbf{z}}^m_{t+1:t+\tau}]$

and use a binary mask $\mathbf{b} \in \{0,1\}^{n+\tau}$ indicating memory slots. Memory slots are treated as clean inputs by forcing their diffusion time to zero, and the mask loss is applied only on future slots. Specifically, we define $\tilde{\mathbf{z}}_s$ as the noised version of the future latent at diffusion level $s$. A compact way to write the noising operation is:

$$\begin{aligned} \mathbf{x}_s &= \mathbf{b} \odot \hat{\mathbf{z}}^o_{t-n+1:t} + (1-\mathbf{b}) \odot \tilde{\mathbf{z}}_s, \\ \tilde{\mathbf{z}}_s &= (1-s)\hat{\mathbf{z}}^m_{t+1:t+\tau} + s\boldsymbol{\epsilon}, \qquad \boldsymbol{\epsilon} \sim \mathcal{N}(\mathbf{0}, \mathbf{I}). \end{aligned} \tag{5}$$

and $\mathcal{L}_{\text{mask}}$ is computed only over $(1-\mathbf{b})$ slots.

**3D RoPE with interpolation scale (compression-aware).** We apply 3D RoPE(Ma et al., 2024) over $(t, h, w)$ token coordinates. Because tokens live in VAE-compressed latent grids, we use an interpolation scale $\boldsymbol{\gamma} = (\gamma_t, \gamma_h, \gamma_w)$ derived from the VAE temporal/spatial compression to keep positional phases consistent across resolutions:

$$\text{RoPE}(t, h, w) \quad \leftarrow \quad \text{RoPE}(\gamma_t\, t, \gamma_h\, h, \gamma_w\, w). \tag{6}$$

We report the concrete setting of $\boldsymbol{\gamma}$ (as a function of the VAE compression ratios and latent frame rate) in the appendix for reproducibility.

**Timestep conditioning via AdaIN-style modulation.** Instead of standard adaptive normalization, MWM applies timestep-dependent scale and shift modulation after normalization(Huang & Belongie, 2017).

For a hidden activation $\mathbf{x}$, we first normalize using RM-SNorm, then modulate:

$$\bar{\mathbf{x}} = \text{RMSNorm}(\mathbf{x}),$$
$$\text{Modulate}(\bar{\mathbf{x}}; s) = \bar{\mathbf{x}} \odot \big(1 + \alpha(s)\big) + \beta(s) \quad (7)$$

Here, $\alpha(s)$ and $\beta(s)$ are functions of the timestep embedding $s$, learned through a combination of a trainable scale-shift table and timestep projection. The operator $\odot$ represents element-wise multiplication.

This design enhances stability when operating on normalized VAE latents by effectively handling their inherent variability, while simultaneously preserving accurate timestep-dependent conditioning for improved performance.

**Predictive feature bank for control.** During mask forecasting, we cache the hidden states from transformer blocks to form a multi-level predictive feature bank $\mathbf{H}_t = \{\mathbf{h}_t^{(1)}, \ldots, \mathbf{h}_t^{(L)}\}$. These features preserve view-/space-/time-indexed structure and are later exposed to the policy head as cross-attention keys/values. We use all blocks by default; aggregation details are deferred to the appendix.

**Multi-view handling (minimal reproducible description).** Multi-view tokens are processed mostly with shared spatiotemporal self-attention, and we periodically apply cross-view mixing layers so that information can flow between views while retaining view-specific structure. The exact cross-view frequency is reported in the appendix.

### 3.4. Mask-Guided Diffusion Policy

**Action-state tokens.** The policy predicts actions in short chunks. At each step, we form an input vector $\mathbf{u}_t = [\mathbf{a}_t, \mathbf{s}_t]$ by concatenating the action parameterization with a low-dimensional proprioceptive state $\mathbf{s}_t$. $\mathbf{u}_t$ is linearly projected into action tokens before entering the action transformer.

**Diffusion objective in action space.** We train the policy as a conditional denoiser in action space. Given an action chunk $\mathbf{u}$, sample $\sigma$ and add noise $\tilde{\mathbf{u}} = \mathbf{u} + \sigma\boldsymbol{\epsilon}$ with $\boldsymbol{\epsilon} \sim \mathcal{N}(\mathbf{0}, \mathbf{I})$. The denoiser $\phi_\xi$ is conditioned on $\mathbf{H}_t$ (and optionally text embeddings) and trained with a weighted score-matching objective:

$$\mathcal{L}_{\text{act}} = \mathbb{E}\Big[\lambda(\sigma)\big\|\phi_\xi(\tilde{\mathbf{u}}, \sigma, \mathbf{H}_t) + \boldsymbol{\epsilon}/\sigma\big\|_2^2\Big]. \quad (8)$$

At test time, we sample an action chunk by iterative denoising and execute in a receding-horizon closed loop.

### 3.5. Two-Stage Training Protocol

**Stage 1: mask dynamics pretraining.** In the first stage, the backbone is trained to predict future semantic mask latents using the flow matching objective ($\mathcal{L}_{\text{mask}}$) defined in Eq. (4), with memory slots fixed by Eq. (5). A memory window of $n{=}4$ RGB frames is used to predict $\tau{=}5$ future latent frames, covering a total of 9 video frames. To enhance robustness, caption dropout ($p{=}0.06$) and slight noise injection (0.1) are applied to the conditioning frames.

**Stage 2: policy learning (no video/mask loss).** Starting from the Stage-1 checkpoint, we optimize the model end-to-end using only the action loss, $\mathcal{L}_{\text{stage2}} = \mathcal{L}_{\text{act}}$. The VAE is frozen, while the DiT backbone and action expert are jointly trained to denoise action chunks ($H_a{=}36$) from backbone features. The action space is 15-dimensional, including a 7-DoF pose, a 1-DoF gripper command, and a 7-DoF state vector. No mask or video reconstruction loss is used in this stage; instead, gradients from $\mathcal{L}_{\text{act}}$ update the backbone so that its predictive features become more control-aligned. We keep the Stage-1 optimizer settings, but use a lower learning rate and disable text dropout. The appendix compares this design against adding an auxiliary mask loss in Stage 2 and against training from scratch without Stage-1 initialization.

**Inference.** During deployment, Receding Horizon Control (RHC) is used to predict a 36-step action chunk via 10-step Euler discrete diffusion sampling. After generating the action chunk, the first action is executed, and the model replans at the next timestep, ensuring adaptive and responsive control.

## 4. Experiments

### 4.1. Simulation Experiments

**Benchmarks.** We evaluate on two standard language-conditioned manipulation benchmarks. LIBERO(Liu et al., 2023) consists of 130 simulated manipulation tasks with templated language instructions; following common practice, we report results on four evaluation suites: LIBERO-Spatial, LIBERO-Object, and LIBERO-Goal (10 tasks each), plus LIBERO-10, a 10-task long-horizon subset derived from LIBERO-100 (Table 1). RLBench(James et al., 2019) contains 100 tabletop manipulation tasks with standardized multi-view observations and natural-language goals; following prior work, we report success rates (SR) on six representative tasks (Table 2). On RLBench, we run 20 evaluation episodes per task with randomized seeds/initializations.

**Baselines and our instantiations.** On LIBERO, we compare against representative RGB-centric generalist policies and world-model pipelines: OpenVLA(Kim et al., 2024), CogACT(Li et al., 2024), $\pi 0$(Black et al., 2026), Cosmos+IDM, Cosmos+LatentIDM, and GE-ACT (Table 1). We additionally evaluate three mask-based instantiations introduced in Section 3 under the *same training data and*

*Table 1.* LIBERO benchmark success rates (SR). For each suite, we select the checkpoint with the best validation performance and evaluate it over 500 episodes. Cell colors indicate camera setups: green for 3rd, yellow for 3rd+wrist, and red for our method.

| Modality | Cameras | Method | Spatial | Object | Goal | Libero-10 | Avg. |
|---|---|---|---|---|---|---|---|
| RGB | 3rd | CogACT | 0.972 | 0.980 | 0.902 | 0.888 | 0.936 |
| | 3rd | OpenVLA | 0.847 | 0.884 | 0.792 | 0.537 | 0.765 |
| | 3rd | Cosmos w/ IDM | 0.768 | 0.750 | 0.694 | 0.488 | 0.675 |
| | 3rd | Cosmos w/ Latent IDM | 0.948 | 0.992 | 0.892 | 0.842 | 0.919 |
| | 3rd+wrist | $\pi$0 | 0.968 | 0.988 | 0.958 | 0.852 | 0.942 |
| | 3rd+wrist | GE-ACT | 0.982 | 0.976 | 0.958 | 0.944 | 0.965 |
| Mask | 3rd | MWM-C1 | 0.866 | 0.842 | 0.828 | 0.704 | 0.810 |
| | 3rd | MWM-C2 | 0.948 | 0.988 | 0.922 | 0.812 | 0.918 |
| | 3rd+wrist | **MWM (ours)** | **0.988** | **1.000** | **0.982** | **0.960** | **0.983** |

*recipe* (detailed architectures are provided in Section C):

- **MWM-C1** explicit future-mask decoding with an IDM

- **MWM-C2** predictive latent features with an action diffusion head, without explicit mask decoding

- **MWM** end-to-end mask world model with third-person + wrist-view inputs

On RLBench, we compare to OpenVLA, CogACT, $\pi$0, FiS-VLA, and GE-ACT.

**Evaluation protocol.** We select checkpoints using a held-out validation split to avoid test-time tuning. For LIBERO, we perform suite-wise selection and evaluate the selected checkpoint over 500 episodes with randomized seeds/initializations. For RLBench, we select the best validation checkpoint per task and evaluate it over 20 episodes. SR is the fraction of successful episodes.

**Results and analysis.** As shown in Table 1, replacing RGB-centric prediction with future semantic structure improves performance consistently across all LIBERO suites. A direct comparison to RGB world-model counterparts shows that mask prediction is particularly beneficial when errors compound over longer horizons: MWM-C1 outperforms Cosmos w/ IDM in average SR (0.675→0.810), with the largest gain on the long-horizon LIBERO-10 suite (0.488→0.704), suggesting reduced rollout drift from appearance-related nuisance variables. Similarly, MWM-C2 improves over Cosmos w/ Latent IDM (0.873→0.918), with clear gains on Goal and LIBERO-10, indicating that learning predictive representations in a semantic space yields more decision-relevant features. Notably, MWM-C2 also surpasses MWM-C1 (0.810→0.918), consistent with the hypothesis that leveraging predictive latent features avoids error propagation introduced by explicitly decoded mask rollouts and a separately trained IDM. With additional wrist-view observations, MWM achieves the best overall performance (0.965→0.983 over GE-ACT) while remaining strong on LIBERO-10, approaching saturation on Spatial/Object/Goal.

On RLBench (Table 2), MWM achieves 68.3% average SR, outperforming both the RGB world-model baseline GE-ACT (30.8%) and the stronger generalist baseline FiS-VLA (50.0%). The largest gains appear on tasks requiring precise object interaction and goal satisfaction (e.g., *Umbrella Out*, *Frame off Hanger*, *Wine at Rack*), supporting the claim that a semantic bottleneck improves robustness to appearance variation and strengthens decision-relevant generalization.

### 4.2. Real-World Experiments

**Setup and tasks.** We evaluate on a Franka Emika Panda robot in a tabletop environment with two synchronized Intel RealSense D435i cameras: a fixed third-person view and an eye-in-hand wrist view. Detailed hardware specifications, sensor layout, and task visualizations are provided in Appendix B. We consider four language-conditioned manipulation tasks: (1) place bread and a hotdog into a basket, (2) open a drawer and place a pen inside, (3) pour water from a bottle into a bowl, and (4) place a book onto a shelf. In Table 4, Task1–Task4 correspond to tasks (1)–(4).

**Data collection, mask annotation, and post-training.** For each task, we collect 50 task-specific demonstrations and perform per-task post-training across all methods to ensure consistent evaluation. To provide semantic supervision for MWM, we utilize RoboEngine (Yuan et al., 2025) to annotate the demonstration videos with pixel-wise semantic masks, covering the robot arm (including the gripper) and task-relevant objects. Notably, RoboEngine is employed only offline to generate training labels, while at test time, MWM operates directly on raw multi-view RGB observations, leveraging its learned mask-centric dynamics for control. For a fair comparison, both GE-ACT and $\pi$0 are post-trained using the same 50 demonstrations per task under identical training settings.

*Table 2.* RLBench success rates. We evaluate the best checkpoint selected on validation over 20 episodes with randomized seeds.

| Method | Sweep to Dustpan | Phone on Base | Umbrella Out | Frame off Hanger | Wine at Rack | Water Plants | Avg. |
|---|---|---|---|---|---|---|---|
| CogACT | 50.0% | 50.0% | 55.0% | 45.0% | 30.0% | 25.0% | 42.5% |
| FiS-VLA | 55.0% | 50.0% | 50.0% | 70.0% | 55.0% | 20.0% | 50.0% |
| OpenVLA | 50.0% | 20.0% | 35.0% | 15.0% | 10.0% | 10.0% | 23.3% |
| $\pi 0$ | 30.0% | 30.0% | 30.0% | 70.0% | 10.0% | 30.0% | 33.3% |
| GE-ACT | 10.0% | 15.0% | 40.0% | 35.0% | 40.0% | 45.0% | 30.8% |
| **MWM (ours)** | **55.0%** | **55.0%** | **85.0%** | **75.0%** | **90.0%** | **50.0%** | **68.3%** |

*Table 3.* Visual generalization on real-world tasks (Tasks (1)–(4)). Each entry is averaged over four tasks, with 20 trials per task per condition. We report in-distribution SR ($\text{SR}_{\text{ID}}$), SR under each appearance shift, and the summary metrics OOD-SR and Retain.

| Method | $\text{SR}_{\text{ID}}$ | BG | Light | Color | OOD-SR | Retain |
|---|---|---|---|---|---|---|
| GE-ACT | 23.8% | 3.8% | 18.8% | 15.0% | 12.5% | 0.53 |
| $\pi 0$ | 38.8% | 13.8% | 17.5% | 26.3% | 19.2% | 0.49 |
| **MWM (ours)** | **67.5%** | **27.5%** | **56.3%** | **42.5%** | **42.1%** | **0.62** |

*Table 4.* Real-world success rates (SR) on a Franka robot. Each method is post-trained per task using 50 demonstrations.

| Method | Task1 | Task2 | Task3 | Task4 | Avg. |
|---|---|---|---|---|---|
| GE-ACT | 35% | 20% | 10% | 30% | 23.8% |
| $\pi 0$ | 50% | 30% | 5% | 70% | 38.8% |
| **MWM (ours)** | **75%** | **55%** | **60%** | **80%** | **67.5%** |

*Table 5.* Robustness to random token pruning on LIBERO (multi-view). We report normalized pruning AUC (nPAUC; higher is better), averaged over pruning ratios $r \in \{0.1, \ldots, 0.9\}$ and the four LIBERO suites. Full detailed are provided in Section D.

| Method | nPAUC ↑ |
|---|---|
| GE-ACT | 0.629 |
| **MWM (ours)** | **0.648** |

**Evaluation protocol.** After post-training, each method is evaluated for 20 real-robot trials per task with randomized initializations/seeds, and we report success rate (SR) as the fraction of successful executions.

**Results and analysis.** Figure 3 visualizes representative real-robot rollouts, and Table 4 reports success rates. Table 4 shows that **MWM** consistently outperforms RGB-centric baselines across all four tasks, achieving 67.5% average SR versus 23.8% for GE-ACT and 38.8% for $\pi 0$. The largest improvements appear on tasks with tighter goal constraints and higher sensitivity to compounding errors, such as drawer manipulation (Task2) and pouring (Task3), where RGB-based policies are more prone to appearance-driven drift. Overall, these results indicate that enforcing a semantic bottleneck together with multi-view observations yields more decision-relevant representations and improves robustness under real-world visual variability, even with only 50 demonstrations per task.

### 4.3. Analysis: Robustness and Generalization

#### 4.3.1. ROBUSTNESS TO RANDOM TOKEN PRUNING

Token pruning/merging is a common way to reduce transformer inference cost by decreasing the number of visual tokens processed by attention. To evaluate robustness under compute-constrained inference, we randomly prune a fraction $r$ of *visual* tokens (uniformly at random) *before* feeding

them into the Video-DiT backbone, while keeping language tokens unchanged. We sweep $r \in R = \{0.1, 0.2, \ldots, 0.9\}$ and report suite-wise success rates $\text{SR}_s(r)$ on LIBERO (500 episodes per suite, following the standard protocol).

**Metric.** To summarize robustness with a single scalar (main paper), we measure the normalized pruning AUC (nPAUC), defined as the average success retention relative to the no-pruning performance: $\text{nPAUC} = \frac{1}{|S||R|} \sum_{s \in S} \sum_{r \in R} \frac{\text{SR}_s(r)}{\text{SR}_s(0)}$, where $S$ denotes the four LIBERO suites and $\text{SR}_s(0)$ is the success rate without pruning. The full $\text{SR}_s(r)$ curves are reported in the appendix.

**Results.** As shown in Table 5, **MWM** achieves a higher nPAUC than **GE-ACT** (0.648 vs. 0.629), indicating slightly better robustness to random token removal. We attribute the gain to the semantic bottleneck: mask-centric tokens concentrate capacity on object geometry and relations, making the policy less sensitive to missing visual tokens than RGB-centric representations.

#### 4.3.2. VISUAL GENERALIZATION

**Protocol.** We evaluate visual generalization on the four real-world tasks in Section 4.2 (Tasks (1)–(4)). All policies are post-trained on 50 demonstrations per task under a *nominal* visual condition, and are then evaluated *without any further adaptation* under controlled appearance shifts. For

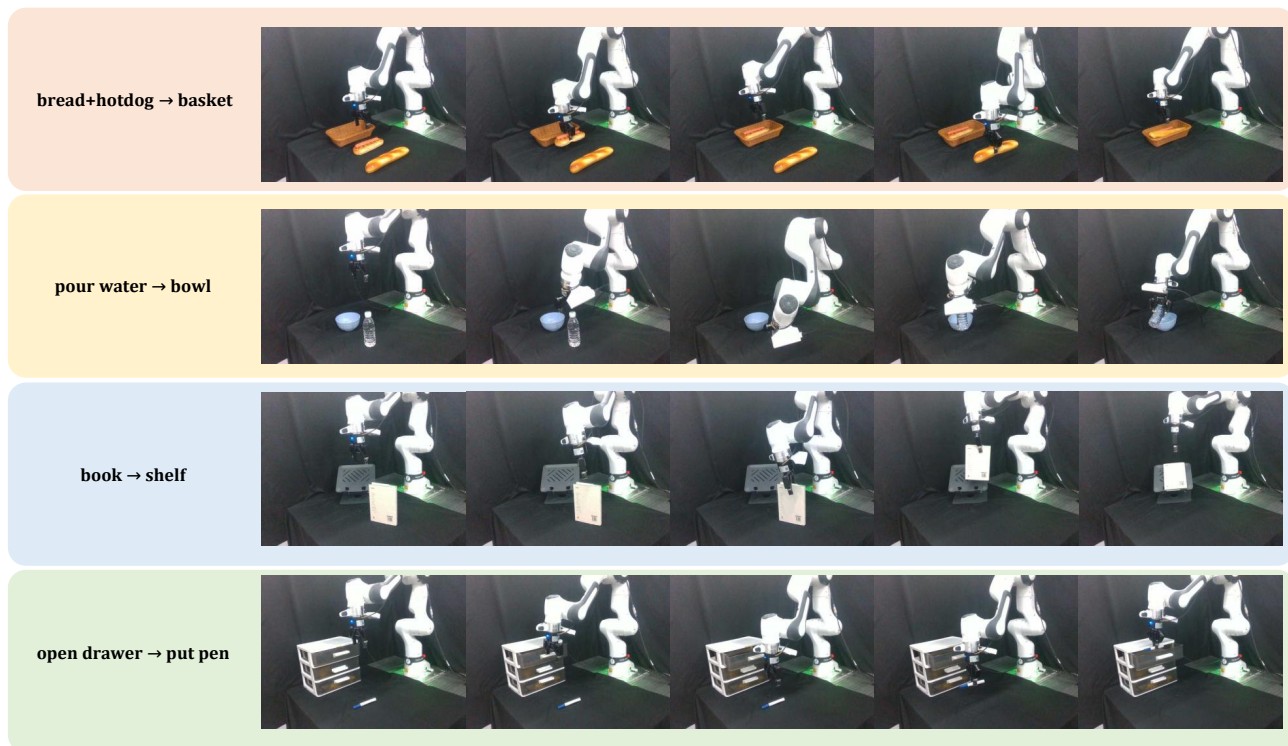

*Figure 3.* **Real-robot qualitative rollouts.** We visualize representative third-person executions for the four real-world tasks: bread+hotdog→basket, pour water→bowl, book→shelf, and open drawer→put pen. Each row shows frames ordered left-to-right.

each task and each shift, we run 20 real-robot trials with randomized initializations and report success rate (SR).

**Appearance shifts.** We consider three appearance factors that commonly induce failures under real-world visual variability: (i) **Background** (BG): replacing the tablecloth with unseen textures/patterns; (ii) **Lighting** (Light): changing illumination intensity (dimmer/brighter) while keeping geometry unchanged; (iii) **Object color** (Color): swapping task objects to unseen colors while preserving shape and size. These shifts primarily perturb visual nuisance variables (texture, illumination, and color) rather than task geometry.

**Metrics.** Let $\mathrm{SR_{ID}}$ denote the in-distribution success rate under the nominal condition, and $\mathrm{SR}_c$ denote the success rate under shift condition $c \in \{\mathrm{BG}, \mathrm{LIGHT}, \mathrm{COLOR}\}$. We summarize the visual robustness using: $\mathrm{OOD\text{-}SR} = \frac{1}{|C|} \sum_{c \in C} \mathrm{SR}_c$, $\mathrm{Retain} = \frac{\mathrm{OOD\text{-}SR}}{\mathrm{SR_{ID}}}$. OOD-SR evaluates SR under appearance shifts, while Retain measures the preservation of nominal performance.

**Results.** Table 3 summarizes OOD performance averaged over the four tasks (20 trials per task per condition; detailed per-task breakdowns are provided in Section E). MWM achieves the highest average performance under appearance shifts (OOD-SR 42.1%), outperforming GE-ACT (12.5%) and $\pi0$ (19.2%). Across shift types, BG is the most dis-ruptive: GE-ACT nearly collapses (3.8% BG-SR), while MWM remains substantially more reliable (27.5%), suggesting reduced sensitivity to background texture cues. Under lighting and color perturbations, MWM maintains strong success (56.3% and 42.5%), indicating improved robustness to illumination and appearance changes. Finally, MWM also attains the best retention of in-distribution performance (0.62), supporting our hypothesis that enforcing a semantic bottleneck yields more decision-relevant representations under real-world visual variability.

## 5. Conclusion

We presented **Mask World Model (MWM)**, a mask-centric world-model framework that mitigates the mismatch between photometric RGB prediction and control utility by forecasting *future semantic masks*. MWM uses semantic supervision only offline during training, while operating purely on raw multi-view RGB at test time, and couples mask-centric predictive features with a diffusion policy head for closed-loop control. Across LIBERO, RLBench, and real-robot evaluations, MWM consistently outperforms strong RGB-centric baselines, demonstrating improved robustness and generalization under real-world visual variability. These results suggest that semantic prediction provides a practical information bottleneck for learning more decision-relevant dynamics representations for generalist robot manipulation.

## Acknowledgements

This work was supported by the National Natural Science Foundation of China (62476011) and (625B2090).

## Impact Statement

This work aims to improve the robustness and reliability of generalist robot manipulation policies by using semantic mask prediction to reduce sensitivity to irrelevant visual variations. The potential positive impacts include more reliable robotic systems in real-world environments and reduced dependence on brittle appearance-based prediction. At the same time, deploying generalist robots may raise broader societal concerns, including safety, labor displacement, and biases introduced by upstream perception or annotation systems. We encourage careful evaluation, transparent reporting, and responsible deployment in safety-critical settings.

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

# A. Implementation Details

We provide a structured overview of the MWM architecture, training protocol, and reproducibility settings. Detailed hyperparameters are summarized in Table 6.

## A.1. Network Architecture

**Video VAE & Tokenization.** We utilize a fixed, pretrained 3D VAE to compress $256 \times 256$ RGB frames into $8 \times 8$ latent patches with channel dimension $D=128$. The compression ratios are $f_s=32$ (spatial) and $f_t=8$ (temporal). To align positional embeddings with this compression, we apply 3D Rotary Positional Embeddings (RoPE) with scaling factors $\gamma = (\gamma_t \approx 0.267, \gamma_h=32, \gamma_w=32)$, compensating for the resolution shift between video and latent space.

**Mask Dynamics Backbone.** The backbone is a 28-layer Diffusion Transformer (DiT) with a hidden dimension of 2048 and 32 attention heads. It processes multi-view inputs (third-person and wrist) via shared spatiotemporal attention. To maintain view consistency while allowing information flow, we insert cross-view attention layers at every $3^{\text{rd}}$ block (indices $0, 3, \ldots, 27$). Text conditioning is injected via a T5-base encoder projecting to 2048 dimensions.

**Action Expert Head.** The policy head is a 28-layer transformer (512 hidden dim, 16 heads) that parallels the backbone. It uses a *Predictive Feature Bank* mechanism: the $l$-th layer of the action transformer attends specifically to the spatial-temporal features extracted from the $l$-th layer of the frozen DiT backbone via cross-attention. This design directly leverages the hierarchical semantic representations learned during mask pretraining.

## A.2. Infrastructure and Reproducibility

**Compute.** Training is performed on a cluster of $8 \times$ NVIDIA A100 (80GB) GPUs using DeepSpeed ZeRO-2. Stage 1 takes $\sim 3.5$ days (30k steps), and Stage 2 takes $\sim 1.5$ days (18k steps) per task suite.

**Data Processing.** Images are resized to $256 \times 256$ and normalized to $[-1, 1]$. Semantic masks are rendered with a consistent color palette before VAE encoding. All experiments use a fixed random seed of 42.

*Table 6.* **Detailed Hyperparameters for MWM.**

| Configuration | Stage 1 (Dynamics) | Stage 2 (Policy) |
|---|---|---|
| *Optimization (AdamW)* | | |
| Learning Rate | $3 \times 10^{-4}$ | $5 \times 10^{-5}$ |
| Batch Size | 128 (global) | 128 (global) |
| Weight Decay | $1 \times 10^{-5}$ | $1 \times 10^{-5}$ |
| Warmup Steps | 1000 | 1000 |
| Gradient Clip | 1.0 | 1.0 |
| Precision | bfloat16 | bfloat16 |
| *Model Architecture* | | |
| Layers | 28 | 28 |
| Hidden Dimension | 2048 | 512 |
| Attention Heads | 32 | 16 |
| Cross-Attn Dim | 2048 | 2048 |
| VAE | Trainable | Frozen |
| Backbone / Policy | Trainable | Trainable |
| *Sequence & Data* | | |
| Context / Horizon | 4 frames / 5 latents | 4 frames / 36 actions |
| Spatial Compression | $f_s = 32$ | - |
| Temporal Compression | $f_t = 8$ | - |
| Resolution | $256 \times 256$ | - |

# B. Real-World Experimental Setup

To supplement the experimental results presented in Section 4.2, we provide a detailed visualization of our real-world hardware setup and task environments in Figure 4.

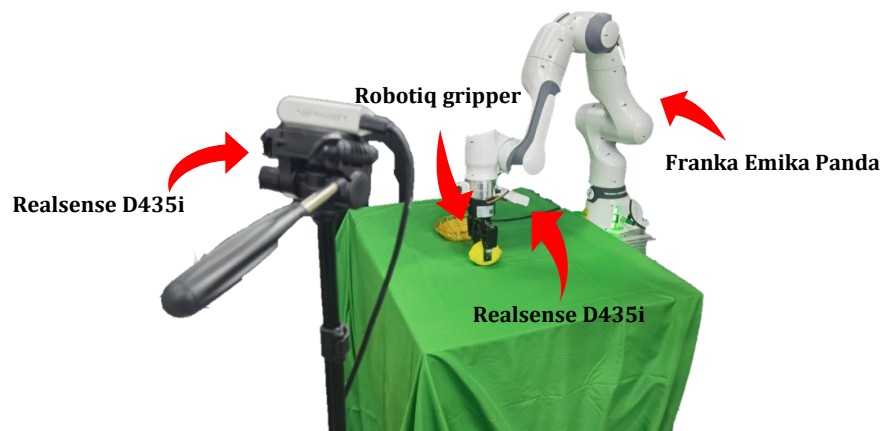

*Figure 4.* **Real-world experimental environment.** (Left) The hardware setup features a Franka Emika Panda robot arm equipped with a parallel gripper. Perception is provided by two synchronized Intel RealSense D435i cameras: a fixed third-person view capturing the global workspace and a wrist-mounted eye-in-hand view for fine-grained interaction. (Right) Snapshots of the four manipulation tasks used for evaluation: (1) Placing food items into a basket, (2) Opening a drawer to insert a pen, (3) Pouring water into a bowl, and (4) Shelving a book. All policies run on raw RGB streams at 10Hz during deployment.

**Hardware Configuration.** Our physical testbed consists of a 7-DoF Franka Emika Panda robot. The RGB-D streams from the two RealSense D435i cameras are resized to $256 \times 256$ resolution. The third-person camera is mounted on a tripod facing the table to cover the entire workspace, while the wrist camera provides ego-centric visual feedback crucial for contact-rich stages of the manipulation tasks.

**Semantic Annotation Process.** For the training of the Mask World Model (Stage 1), we require semantic segmentation masks corresponding to the video demonstrations. We utilize **RoboEngine** (Yuan et al., 2025) to automatically generate pixel-wise semantic annotations for the robot arm, gripper, and task-relevant objects (e.g., bread, drawer handle, cup) offline. These masks serve strictly as supervision targets during the pre-training phase; at inference time, MWM relies solely on the raw multi-view RGB inputs.

## C. Detailed Algorithms of MWM Variants

We provide the detailed training and inference procedures for MWM-C1 and MWM-C2 in Algorithm 1 and Algorithm 2, respectively.

## D. Detailed Random Token Pruning Results

In this section, we report the detailed success rates for the Random Token Pruning (RTP) stress test on LIBERO. We compare **MWM (Ours)** and **GE-ACT** across pruning ratios $r \in \{0.1, \ldots, 0.9\}$ on four evaluation suites: Spatial, Object, Goal, and Libero-10.

## E. Detailed Visual Generalization Results

In this section, we provide the per-task success rates for the visual generalization experiments discussed in Section 4.3.2. We evaluated three methods (GE-ACT, $\pi 0$, and our MWM) across four real-world tasks under three distinct appearance shifts: Background (BG), Lighting (Light), and Object Color (Color).

To provide qualitative context for these quantitative results, we visualize the specific environmental variations in Figure 5. These shifts serve as stress tests to determine whether the models are overfitting to visual nuisance variables (e.g., table texture or illumination) rather than learning the underlying manipulation geometry.

**Quantitative Results.** The detailed success rates for each shift type are presented below. Table 9 details the performance under background changes, Table 10 covers lighting variations, and Table 11 reports robustness to object color swaps.

*Table 7.* **MWM (Ours) Robustness Profile.** Success rates at varying visual token pruning ratios.

| Pruning Ratio ($r$) | LIBERO-Spatial | LIBERO-Object | LIBERO-Goal | LIBERO-10 |
|:---:|:---:|:---:|:---:|:---:|
| 0.1 | 0.98 | 1.00 | 0.98 | 0.96 |
| 0.2 | 0.97 | 1.00 | 0.97 | 0.96 |
| 0.3 | 0.96 | 0.99 | 0.94 | 0.95 |
| 0.4 | 0.95 | 0.97 | 0.89 | 0.79 |
| 0.5 | 0.90 | 0.94 | 0.84 | 0.69 |
| 0.6 | 0.85 | 0.80 | 0.65 | 0.37 |
| 0.7 | 0.50 | 0.49 | 0.40 | 0.03 |
| 0.8 | 0.05 | 0.10 | 0.07 | 0.00 |
| 0.9 | 0.00 | 0.00 | 0.00 | 0.00 |

*Table 8.* **GE-ACT Robustness Profile.** Success rates at varying visual token pruning ratios.

| Pruning Ratio ($r$) | LIBERO-Spatial | LIBERO-Object | LIBERO-Goal | LIBERO-10 |
|:---:|:---:|:---:|:---:|:---:|
| 0.1 | 0.96 | 0.99 | 0.95 | 0.94 |
| 0.2 | 0.96 | 0.98 | 0.94 | 0.95 |
| 0.3 | 0.96 | 0.97 | 0.94 | 0.86 |
| 0.4 | 0.96 | 0.96 | 0.89 | 0.77 |
| 0.5 | 0.90 | 0.85 | 0.84 | 0.59 |
| 0.6 | 0.83 | 0.47 | 0.67 | 0.39 |
| 0.7 | 0.51 | 0.09 | 0.42 | 0.03 |
| 0.8 | 0.11 | 0.00 | 0.21 | 0.00 |
| 0.9 | 0.00 | 0.00 | 0.00 | 0.00 |

*Table 9.* **Detailed success rates under Background (BG) shift.** The tablecloth texture was replaced with unseen patterns. Reported values are success rates (%) over 20 trials per task.

| METHOD | TASK1 | TASK2 | TASK3 | TASK4 | AVG. |
|:---|:---:|:---:|:---:|:---:|:---:|
| GE-ACT | 0% | 0% | 10% | 5% | 3.8% |
| $\pi 0$ | 10% | 20% | 15% | 10% | 13.8% |
| **MWM (OURS)** | **35%** | **25%** | **20%** | **30%** | **27.5%** |

*Table 10.* **Detailed success rates under Lighting (Light) shift.** Illumination intensity was significantly altered (dimmer/brighter). Reported values are success rates (%) over 20 trials per task.

| METHOD | TASK1 | TASK2 | TASK3 | TASK4 | AVG. |
|:---|:---:|:---:|:---:|:---:|:---:|
| GE-ACT | 30% | 15% | 10% | 20% | 18.8% |
| $\pi 0$ | 30% | 20% | 0% | 20% | 17.5% |
| **MWM (OURS)** | **65%** | **50%** | **50%** | **60%** | **56.3%** |

*Table 11.* **Detailed success rates under Object Color (Color) shift.** Task objects were swapped with instances of unseen colors. Reported values are success rates (%) over 20 trials per task.

| METHOD | TASK1 | TASK2 | TASK3 | TASK4 | AVG. |
|:---|:---:|:---:|:---:|:---:|:---:|
| GE-ACT | 25% | 5% | 10% | 20% | 15.0% |
| $\pi 0$ | 30% | 10% | 0% | 65% | 26.3% |
| **MWM (OURS)** | **45%** | **25%** | **50%** | **50%** | **42.5%** |

---

**Algorithm 1** MWM-C1: Cosmos-Predict2 with Inverse Dynamics

---

1: **Input:** Observation $o_t$, language instruction $l$, ground truth masks $m$, actions $a$
2: **Output:** Trained models $\mathcal{W}_\theta, \phi_\psi$ and inference result $\hat{a}$
3: **Step 1: Train Mask World Model $\mathcal{W}_\theta$**
4: **while** not converged **do**
5:     Sample video batch: $v \leftarrow \text{Sample}(\mathcal{D})$;
6:     Convert to mask latents: $z_m \leftarrow \mathcal{E}(v_{mask})$;
7:     Compute diffusion loss: $\mathcal{L}_{\text{cosmos}} \leftarrow \|\epsilon_\theta(z_m + \sigma\epsilon, t, c) - \epsilon\|^2$;
8:     Update $\theta \leftarrow \theta - \eta\nabla_\theta \mathcal{L}_{\text{cosmos}}$;
9: **end while**
10: **Step 2: Train Inverse Dynamics Model (IDM) $\phi_\psi$**
11: **while** not converged **do**
12:     Sample adjacent masks: $(m_t, m_{t+1}, a_t) \sim \mathcal{D}$;
13:     Predict action: $\hat{a}_t \leftarrow \phi_\psi(m_t, m_{t+1})$;
14:     Update $\psi \leftarrow \psi - \eta\nabla_\psi \|\hat{a}_t - a_t\|^2$;
15: **end while**
16: **Step 3: Inference Pipeline**
17: Given current observation $o_t$ and instruction $l$:
18: Generate future mask rollout: $\hat{m}_{t+1:t+\tau} \leftarrow \text{Denoise}(\mathcal{W}_\theta, o_t, l)$;
19: **for** $k = 0$ to $\tau - 1$ **do**
20:     Infer action: $\hat{a}_{t+k} \leftarrow \phi_\psi(\hat{m}_{t+k}, \hat{m}_{t+k+1})$;
21: **end for**
22: **Return** Action sequence $\hat{a}_{t:t+\tau}$

---

---

**Algorithm 2** MWM-C2: Predictive Feature Policy

---

1: **Input:** Observation $o_t$, language instruction $l$, ground truth actions $a$
2: **Output:** Trained policy $\pi_\xi$ and inference result $\hat{a}$
3: **Step 1: Pretrain and Freeze Backbone**
4: Pretrain World Model $\mathcal{W}_\theta$ on mask prediction (same as C1, Step 1);
5: Freeze weights: $\theta \leftarrow \text{StopGradient}(\theta)$;
6: **Step 2: Train Diffusion Policy Head $\pi_\xi$**
7: **while** not converged **do**
8:     Extract predictive features: $\mathbf{F} \leftarrow \mathcal{W}_\theta(o_t, l)$;
9:     // $\mathbf{F}$ contains semantic-aware spatio-temporal tokens
10:     Sample noise $\epsilon$ and timestep $k$;
11:     Noised action: $\tilde{a} \leftarrow a + \sigma_k\epsilon$;
12:     Predict noise: $\hat{\epsilon} \leftarrow \pi_\xi(\tilde{a}, k, \mathbf{F})$;
13:     Update $\xi \leftarrow \xi - \eta\nabla_\xi \|\hat{\epsilon} - \epsilon\|^2$;
14: **end while**
15: **Step 3: Inference Pipeline**
16: Given current observation $o_t$:
17: Extract features (single forward pass): $\mathbf{F} \leftarrow \mathcal{W}_\theta(o_t, l)$;
18: Initialize noise action: $\hat{a}_K \sim \mathcal{N}(0, I)$;
19: **for** $k = K$ down to 1 **do**
20:     Denoise action: $\hat{a}_{k-1} \leftarrow \text{Sampler}(\pi_\xi, \hat{a}_k, \mathbf{F})$;
21: **end for**
22: **Return** Denoised action $\hat{a}_0$

---

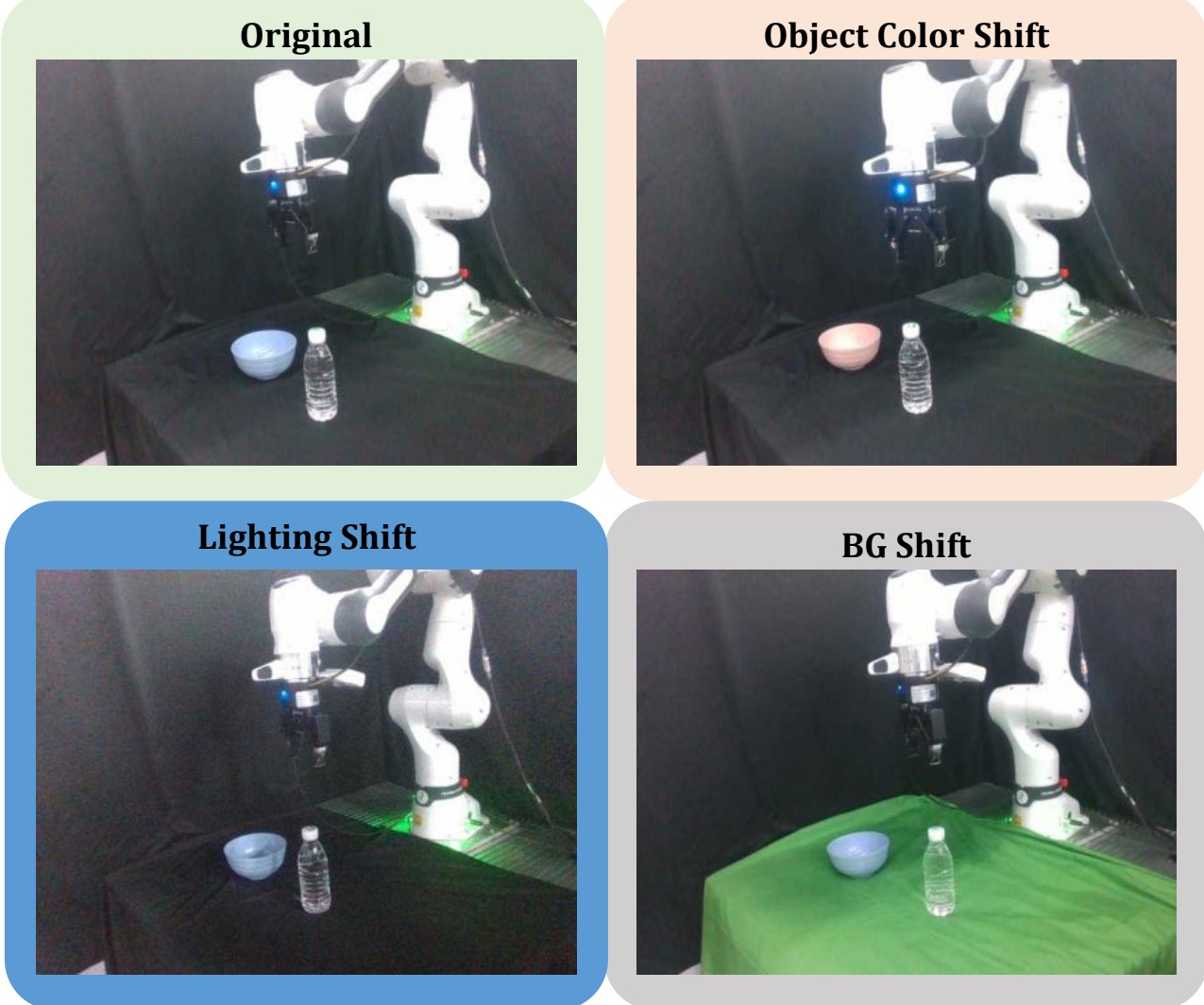

*Figure 5.* **Visual generalization stress tests.** We evaluate policy robustness under three distinct distribution shifts relative to the nominal condition (Top-Left). The shifts include: (Top-Right) **Object Color Shift**, where task objects are swapped with unseen colors while retaining geometry; (Bottom-Left) **Lighting Shift**, involving significant changes in illumination intensity; and (Bottom-Right) **Background (BG) Shift**, where the table surface is replaced with unseen textures. These setups correspond to the quantitative results reported in Tables 9–11.

