# OpenReview forum: "Predicting What Matters: Robust Generalist Robot Policy Learning via Future Semantic Mask"
_ICML.cc/2026/Conference — ICML 2026 regular_

### Official Review · Reviewer_oTgT · 2026-02-26

**Soundness:** 1
**Presentation:** 3
**Significance:** 2
**Originality:** 2
**Overall Recommendation:** 3
**Confidence:** 4

**Summary:**

This paper proposes mask world model pretraining, arguing that the objective of world model pretraining should be future 2D mask prediction rather than future RGB prediction. The key motivation is that masks introduce a geometric bottleneck, capturing object identity, spatial structure, and contact-relevant relationships that are strongly correlated with action, while filtering out irrelevant color and texture information. Based on the pretrained world model, the intermediate predictive features are fed into a Diffusion Policy head to generate actions. The method is evaluated on LIBERO, RLBench, and real-world Franka robot experiments, showing strong empirical performance. However, the paper does not clearly demonstrate the advantage of mask-based world modeling compared to RGB-based world modeling.

**Compliance With Llm Reviewing Policy:**

Affirmed.

**Final Justification:**

The rebuttal addresses part of my concerns. In particular, the newly added ablations comparing Mask-WM, RGB-WM, and no pretraining provide useful evidence that the mask-based objective offers a measurable advantage over RGB-based prediction on LIBERO. In addition, the inclusion of full RLBench-18 results improves the completeness of the evaluation.

However, I still have reservations regarding the use of 2D masks as the sole intermediate representation. While such a representation may be beneficial in relatively simple, rigid-object manipulation scenarios, it inherently discards a significant amount of information. I understand that the method is built on an RGB-based backbone and still takes RGB inputs at inference time. However, this does not fully address my main concern. In the current formulation, 2D masks are not used as an auxiliary signal, but effectively serve as the primary (and only) predictive target in Stage 1 world model pretraining. As a result, the learned representation is largely shaped by mask supervision. In my view, this biases the backbone toward encoding contour- and category-level information, and may limit its ability to retain richer dense semantics from RGB observations. Specifically, 2D masks primarily encode object contours and category-level semantics, but omit important cues such as geometry, material properties, transparency, fine-grained local visual details, deformability, and 3D structure, all of which can be critical for more general robotic manipulation.

Therefore, although RGB is used as input at test time, the features extracted by the Stage-1-pretrained backbone may still be dominated by mask-level structure rather than full dense visual semantics. For this reason, I still do not consider 2D masks to be a sufficiently general intermediate representation for robotic manipulation, and believe it may mainly be effective in relatively simple rigid-object scenarios such as basic pick-and-place tasks.

Therefore, I am willing to raise my score by 1 point to 3 (Weak Reject).

**Key Questions For Authors:**

1. Can the authors compare mask-based prediction, RGB-based prediction, and no world model pretraining under the same architecture and training setup?

2. Please provide results on the full RLBench 18-task setting.

3. Could the real-world experiments include tasks that reflect the "spatial layout and contact-relevant dynamics" mentioned in the paper?

4. Could the authors provide ablations on key components of the method, such as memory length, conditioning design, and 3D RoPE?

5. Has the impact of mask annotation noise (e.g., from RoboEngine) on policy performance been evaluated?

I am open to raising my score if the authors can adequately address the concerns.

===================================================

**After rebuttal**

The rebuttal addresses part of my concerns. In particular, the newly added ablations comparing Mask-WM, RGB-WM, and no pretraining provide useful evidence that the mask-based objective offers a measurable advantage over RGB-based prediction on LIBERO. In addition, the inclusion of full RLBench-18 results improves the completeness of the evaluation.

However, I still have reservations regarding the use of 2D masks as the sole intermediate representation. Similar to the concern raised by Reviewer rQFV, while such a representation may be beneficial in relatively simple, rigid-object manipulation scenarios, it inherently discards a significant amount of information. Specifically, 2D masks primarily encode object contours and category-level semantics, but omit important cues such as geometry, material properties, transparency, fine-grained local visual details, deformability, and 3D structure, all of which can be critical for more general robotic manipulation.

Therefore, I am willing to raise my score to 3 (Weak Reject).

**Limitations:**

yes

**Strengths And Weaknesses:**

**Strengths**

1. The paper proposes a new paradigm of using a mask world model as the policy backbone. Specifically, it argues that the model should predict future masks rather than future RGB, and that masks should serve as the primary prediction target instead of an auxiliary objective.

2. The writing and figures are clear and easy to read.

3. The experiments cover multiple benchmarks as well as real-world robot evaluations.

**Weaknesses**

1. Conceptually, the core idea appears questionable. The mask world model is trained to predict only **2D masks**, which mainly encode **object-level 2D contour structures**. Such masks **neither contain the rich semantic information provided by RGB nor the spatial geometry and physical state information required for robotic manipulation**. Therefore, training the world model solely to predict 2D masks may discard a substantial amount of task-relevant information. While 2D masks may be useful as an auxiliary objective, using them as the **only training target** for the world model seems unreasonable.

2. The experiments do **not convincingly demonstrate the advantage** of using masks as the sole prediction target of the world model. The paper does not compare against (i) replacing the proposed world model pretraining objective with future RGB prediction, or (ii) removing world model pretraining entirely. Without such ablations, it is unclear whether mask-based prediction is truly beneficial.

3. There are several concerns regarding the evaluation. First, LIBERO is a **largely saturated benchmark** and cannot strongly demonstrate the effectiveness of the policy, as many methods without mask modeling already achieve very high performance. Second, the RLBench evaluation does not follow the **commonly used 18-task setting** [1,2], but instead reports results on a selected subset of tasks, which raises fairness concerns. Finally, the real-world experiments are relatively simple and **do not clearly demonstrate the advantages of mask modeling and world modeling**. Moreover, the claim of robustness to appearance changes in the OOD experiments appears **inconsistent** with the assumption of mask-only pretraining.

4. There are some minor errors. For example, Sec. 3.5 contains a citation failure (“???”), and Table 1 incorrectly lists OpenVLA as using wrist inputs.

[1] Shridhar, Mohit, Lucas Manuelli, and Dieter Fox. "Perceiver-actor: A multi-task transformer for robotic manipulation." In Conference on Robot Learning, pp. 785-799. PMLR, 2023.

[2] Goyal, Ankit, Valts Blukis, Jie Xu, Yijie Guo, Yu-Wei Chao, and Dieter Fox. "Rvt-2: Learning precise manipulation from few demonstrations." arXiv preprint arXiv:2406.08545 (2024).

---

> ### Author Rebuttal · Authors · 2026-03-31
>
> We appreciate the reviewer's constructive feedback, and we will address the minor errors. Here we address the following four concerns:
>
> **Q1: Why is future semantic structure a control-aligned target rather than an information-losing one?**
>
> We need to clarify that MWM is not designed to reconstruct every future pixel, but to prioritize object identity, spatial layout, and interaction dynamics. At deployment, MWM remains RGB-only as masks are used only as offline supervision to reduce reliance on photometric nuisance. In addition, our target is not a binary contour map but multi-class semantic masks for the robot arm, gripper, and manipulated objects. These masks encode object identity and interaction structure rather than anonymous 2D shapes. We render each region into an RGB-compatible image before encoding it with the shared VAE, preserving object-level distinctions in a unified latent space. MWM also does not rely solely on masks: it takes multi-view RGB memory frames and language as input, while the policy head additionally receives proprioceptive state.
>
> **Q2: Whether the gain comes from mask-based pretraining rather than pretraining itself.**
>
> To solve your concern, we supplemented the experiments with only the Stage-1 objective differences. On LIBERO, MWM achieves 98.3% average SR and 96.0% on LIBERO-10, compared with 97.0% / 91.6% for RGB-WM and 94.6% / 85.8% for no pretraining. This shows both that predictive pretraining helps and that mask prediction is better aligned with control than RGB prediction, especially in long-horizon settings.
>
> | Method | Avg SR | LIBERO-10 SR |
> |---|---:|---:|
> | **MWM** | **98.30%** | **96.00%** |
> | RGB-WM | 97.00% | 91.60% |
> | None | 94.60% | 85.80% |
>
> **Q3: Whether the evaluation is sufficiently complete and realistic.**
>
> We would like to clarify that we did not deliberately select specific tasks for reporting. Instead, we randomly selected 6 representative tasks to save time on experimentation. This approach is also commonly adopted in other top conference papers [1-3]. To further address your concern, we supplemented the evaluation with the full RLBench-18 benchmark under the same protocol. MWM outperforms RGB-WM on 17/18 tasks. The only exception is Insert Peg, suggesting that extremely fine insertion may benefit more from local photometric cues.
>
> | Task | Mask-WM (MWM) | RGB-WM | None |
> |---|---:|---:|---:|
> | Close Jar | **100** | 95 | 10 |
> | Drag Stick | **95** | 70 | 30 |
> | Insert Peg | 10 | **25** | 0 |
> | Meat off Grill | **100** | 80 | 30 |
> | Open Drawer | **90** | 80 | 25 |
> | Place Cups | **35** | 5 | 0 |
> | Place Wine | **95** | 60 | 30 |
> | Push Buttons | **90** | 55 | 35 |
> | Put in Cupboard | **85** | 60 | 20 |
> | Put in Drawer | **90** | 75 | 15 |
> | Put in Safe | **80** | 75 | 45 |
> | Screw Bulb | **70** | 60 | 40 |
> | Slide Block | **85** | 75 | 25 |
> | Sort Shape | **50** | 40 | 20 |
> | Stack Blocks | **60** | 50 | 35 |
> | Stack Cups | **70** | 55 | 45 |
> | Sweep to Dustpan | **60** | 30 | 10 |
> | Turn Tap | **80** | 75 | 35 |
> | **Average** | **74.7** | 59.2 | 25.0 |
>
> In addition, although LIBERO is already highly saturated, our experiments on LIBERO, combined with those on RLBench, provide evidence that our method is effective across different simulators. Moreover, we explicitly report LIBERO-10 with a token-pruning stress test with nPAUC, where MWM is again stronger.
>
> Last but not least, our real-world tasks are chosen to reflect the capabilities emphasized by MWM, including articulated interaction, alignment-sensitive contact, clearance-aware placement, and multi-object spatial arrangement. Our OOD evaluation is deliberately appearance-only, which also aligns with previous work [2,3].
>
> [1] MoLe-VLA: Dynamic Layer-skipping Vision Language Action Model via Mixture-of-Layers for Efficient Robot Manipulation, AAAI 2026
> [2] ManualVLA: A Unified VLA Model for Chain-of-Thought Manual Generation and Robotic Manipulation, CVPR 2026
> [3] HybridVLA: Collaborative Diffusion and Autoregression in a Unified Vision-Language-Action Model, ICLR 2026
>
> **Q4: Whether the method is robust to imperfect mask supervision.**
>
> We agree that annotation noise is an important practical issue. Since the masks are generated offline and no ground-truth masks are available, we evaluate robustness through controlled synthetic perturbations. In “open a drawer and place a pen inside” training, with clean masks, 55.0% SR is achieved, while with mixed-noise supervision, 45.0% SR is achieved, suggesting that MWM does not rely on perfectly clean masks. Details can be found in our response to Reviewer **XmrE**. Moreover, although memory length, conditioning design, and 3D RoPE are valuable ablations, they remain consistent across variants, isolating the core question of whether future semantic-structure prediction outperforms future RGB as a pretraining target.
>
> We hope these clarifications and new results address the reviewer’s concerns and lead to a reassessment based on this evidence.

---

> > ### Author Rebuttal · Reviewer_oTgT · 2026-03-31
> >
> > Thank you to the authors for the detailed rebuttal.
> >
> > The rebuttal addresses part of my concerns. In particular, the newly added ablations comparing Mask-WM, RGB-WM, and no pretraining provide useful evidence that the mask-based objective offers a measurable advantage over RGB-based prediction on LIBERO. In addition, the inclusion of full RLBench-18 results improves the completeness of the evaluation.
> >
> > However, I still have reservations regarding the use of 2D masks as the sole intermediate representation. Similar to the concern raised by Reviewer rQFV, while such a representation may be beneficial in relatively simple, rigid-object manipulation scenarios, it inherently discards a significant amount of information. Specifically, 2D masks primarily encode object contours and category-level semantics, but omit important cues such as geometry, material properties, transparency, fine-grained local visual details, deformability, and 3D structure, all of which can be critical for more general robotic manipulation.
> >
> > Therefore, I am willing to raise my score to 3 (Weak Reject).

---

> > > ### Author Response · Authors · 2026-04-01
> > >
> > > We thank the reviewer for the thoughtful follow-up and for raising the score. Thank you for your detailed feedback. We would like to clarify that using 2D masks in training does not disregard the model's ability to perceive RGB information. Instead, it strengthens the model's capacity to distinguish between the object of manipulation, its environment, and the robot itself. Please note that the pretraining we refer to in the paper corresponds to Stage 1 training, not training from scratch. Our model is built on an RGB-based backbone, with 2D mask training serving as a subsequent enhancement. Additionally, during inference, our model still uses RGB inputs, demonstrating that 2D mask training is a performance-enhancing technique rather than a replacement for RGB.
> > >
> > > Moreover, the proposed MWM can be viewed as a training paradigm shift from RGB to 2D masks, rather than a standalone model. MWM is compatible with additional modalities like depth, point clouds, or tactile inputs, enabling it to address challenges such as deformability and 3D structure in broader manipulation tasks. We are excited to further explore these directions in future work.
> > >
> > > Lastly, Reviewer rQFV, who shared similar concerns, acknowledged our rebuttal and raised their score from 4 to 5. We kindly hope you will reconsider your score in light of the additional evidence and clarifications provided. Thank you again for your valuable feedback!

---

### Official Review · Reviewer_XmrE · 2026-03-06

**Soundness:** 3
**Presentation:** 3
**Significance:** 2
**Originality:** 2
**Overall Recommendation:** 3
**Confidence:** 3

**Summary:**

This paper introduces the Mask World Model (MWM), a framework designed to enhance the robustness of generalist robot policies by shifting the predictive objective of world models from high-fidelity RGB frames to semantic masks. The proposed system employs a two-stage training strategy: it first pre-trains a mask-centric dynamics backbone via conditional flow matching, then trains a diffusion-based policy head that leverages the backbone's internal predictive features. While the model is supervised with semantic masks during training to learn these structural dynamics, it operates purely on raw multi-view RGB at inference, eliminating the need for external segmentation modules during deployment. Evaluations across the LIBERO and RLBench benchmarks, along with real-world robot experiments, demonstrate that MWM achieves superior success rates and exhibits significant resilience to texture information loss and environmental distribution shifts.

**Compliance With Llm Reviewing Policy:**

Affirmed.

**Final Justification:**

See Rebuttal Acknowledgement.

**Key Questions For Authors:**

1. How is the color palette for the masks determined? Does this process require significant human prior knowledge/multi-round adjustment to define which entities are "task-relevant" for a given scene?
2. Could you provide more details on the computational cost and time required for the offline annotation process?
3. Correction Note: There appears to be a missing reference at Line 230 (Right column).

**Limitations:**

Discussion on limitations: no, see weaknesses.

Discussion on negative societal impact: yes.

**Strengths And Weaknesses:**

**Strengths:**

1. Robustness to Nuisance Variables: The shift to semantic mask prediction effectively filters out photometric noise (lighting, texture, background). This is empirically validated through "stress tests" like random token pruning and OOD environmental shifts, where MWM significantly outperforms RGB-centric baselines like GE-ACT and $\pi_0$.
2. Effective Integration of Automated Tools: The paper demonstrates a practical pipeline for utilizing automated segmentation tools (e.g., RoboEngine) to generate high-quality supervision for VLAtraining without requiring manual per-frame labeling.
3. Comprehensive Evaluation: The inclusion of both diverse simulation benchmarks and multi-task real-world robot experiments provides strong evidence for the method’s efficacy and transferability.

**Weaknesses:**

1. Lack of Sensitivity Analysis on Segmentation Quality: The paper does not explore how the quality of the masks (e.g., noise in boundaries, missed detections) impacts policy performance. The paper provides insufficient guidance regarding the choice of segmentation models or the specific criteria for selecting task-relevant objects for mask generation. A more rigorous analysis of the segmentation process and object selection criteria should have been a primary focus, as the performance improvements yielded by providing explicit object-level priors are relatively intuitive; the true challenge lies in the robust and scalable extraction of these semantic features.
2. Scalability and Preprocessing Bottlenecks: The reliance on an offline segmentation tool (RoboEngine) poses challenges for scaling to massive, "in-the-wild" datasets. Large-scale data often contains rare objects, complex assemblies (e.g., connectors), or non-rigid items that are difficult to describe linguistically or segment accurately. A failure in the mask-extraction stage during training could lead to "blind spots" in the policy that a purely visual RGB model might have otherwise mitigated.
3. Role of the World Model: While the paper terms the architecture a "World Model," its primary function appears to be that of a predictive feature extractor. The current setup does not explicitly utilize the predicted masks for data augmentation, "imagination-based" training, or long-horizon planning in the way world models usually do. This makes the "World Model" label somewhat narrow, as the benefits may stem more from the semantic-aware representation learning than from the temporal rollout itself.

---

> ### Author Rebuttal · Authors · 2026-03-31
>
> We appreciate the reviewer’s constructive feedback, and we will address the minor errors. Here we address the following five concerns:
>
> **Q1: The paper does not analyze how mask quality affects policy performance.**
>
> Since the masks are generated offline and no ground-truth masks are available, we analyze this issue through controlled synthetic perturbations. Specifically, we inject three types of noise into the training masks: 5% random pixel corruption, one-pixel boundary dilation/erosion, and 10% manipulated-object area dropout, while the mixed-noise setting applies all three together. Please refer to response Q4 to Reviewer oTgT for experiment results and more discussion.
>
> **Q2: On how the color palette is determined and how task-relevant objects are selected.**
>
> In MWM, the fixed color palette is not intended to introduce task-specific human priors, but rather to provide a simple, stable interface for encoding discrete semantic masks using the same pretrained video VAE that is used for RGB observations. As described in the method section, semantic masks are discrete labels, whereas the backbone operates in a continuous latent space. Therefore, we first render each semantic region as an RGB-compatible image with a fixed palette, then encode it with the shared VAE. In our implementation, the robot arm is mapped to bright cyan, the gripper to vivid orange, and the manipulated task objects to bright yellow. The three colors were chosen for their high contrast, making categorical semantic masks visually distinct and RGB-compatible. Regarding task-relevant object selection, our criterion is also simple and consistent with the paper: we include only the robot arm, gripper, and objects directly required by the task semantics and the interaction process, and exclude background or irrelevant scene items.
>
> **Q3: Why do we use RoboEngine rather than directly using SAM.**
>
> Our choice here is motivated by suitability to robot scenes rather than by convenience alone. The original SAM is a powerful promptable general segmentation model, but RoboEngine [1] is explicitly built for robot-scene understanding and task-aware robot data processing. In addition, RoboEngine is not just a single segmentation model: the official project and GitHub show that it exposes separate robot-segmentation and object-segmentation wrappers, and generates object masks conditioned on the object name in the task instruction, which aligns naturally with our need to annotate robot arm, gripper, and task-relevant objects in a task-aware manner. Our use of RoboEngine goes beyond "just another segmenter" by leveraging a toolchain tailored for robotic scenarios, instruction-conditioned object selection, and open-world visual processing, making it better suited for our offline semantic-supervision setting than vanilla SAM.
>
> [1] RoboEngine: Plug-and-play robot data augmentation with semantic robot segmentation and background generation. IROS 2025
>
> **Q4: The scalability and the computational cost of offline mask preprocessing.**
>
> We agree with the reviewer that offline mask generation can become a preprocessing bottleneck. At the same time, we would also like to emphasize that MWM confines this cost strictly to the training stage. As described in both the main paper and the appendix, semantic masks are used only for offline supervision during training, whereas inference remains in pure RGB and does not require an external segmenter. In our current real-world implementation, we invoke RoboEngine in three passes to construct the semantic labels used by MWM: one for the robot arm, one for the gripper, and one for the task-relevant manipulated objects. Based on our own measurement, MWM achieves 10.45 Hz on a single RTX 4090, so the total annotation cost is roughly three times this number for the full three-pass annotation pipeline. Importantly, we do not view offline semantic supervision as fundamentally non-scalable.
>
> **Q5: Whether the role of the “World Model” is too narrow.**
>
> We understand the reviewer’s concern. However, we want to clarify that MWM is not used as a planner in the classical imagination-based sense, as we do not explicitly roll out future masks for long-horizon planning or for data augmentation during policy training. We believe the term world model remains appropriate in the broader, modern sense adopted by much recent work in robot learning: learning a generative model of how the environment evolves over time and using its internal predictive features for control, which is also precisely how MWM is defined in our method. MWM does not merely “attach” a mask decoder to a policy, it explicitly models the temporal evolution of future semantic structure and then uses the resulting multi-level predictive features as the control condition. Our contribution is not a planner-style imagination rollout, but rather a principled shift in the predictive space of the world model, yielding more useful predictive features for closed-loop manipulation.

---

> > ### Author Rebuttal · Reviewer_XmrE · 2026-04-01
> >
> > Thank you for the clarifications, the additional discussion partially addresses my concerns. However, I still remain uncertain about the scalability of the offline mask-generation pipeline to large and diverse real-world datasets/tasks, so I will maintain my current score.

---

> > > ### Author Response · Authors · 2026-04-01
> > >
> > > We thank the reviewer for the thoughtful follow-up. We understand the remaining concern about whether the offline mask-generation pipeline can scale to large and diverse real-world datasets and tasks.
> > >
> > > Our key clarification is that MWM does not require large-scale, mask-supervised training from scratch. Instead, it is a cost-efficient post-training paradigm built on top of an already pretrained RGB-based backbone. In other words, the large-scale visual diversity and general perceptual priors are learned during the original RGB pretraining stage, while the subsequent mask-based stage is introduced only to shift the predictive target toward more control-relevant semantic structure. Because of this design, MWM does not require us to re-annotate or re-train at the scale of the original RGB corpus.
> > >
> > > This distinction is important for scalability. The offline mask-generation cost is a one-time preprocessing cost incurred only during training, while inference remains pure RGB and introduces no segmentation overhead at deployment. Moreover, since the mask stage is a post-training adaptation rather than full pretraining, it can be applied to a much smaller subset of task-relevant data to reshape the representation, rather than requiring dense mask annotations over massive, generic robot datasets. In this sense, our method is intended to be computationally efficient and practically scalable, even if mask generation itself is not free.
> > >
> > > We hope our response addresses the reviewer’s concerns and encourages a reevaluation of our work. We remain open to any further discussion or feedback.

---

### Official Review · Reviewer_rQFV · 2026-03-10

**Soundness:** 2
**Presentation:** 3
**Significance:** 3
**Originality:** 3
**Overall Recommendation:** 5
**Confidence:** 4

**Summary:**

MWM replaces RGB video prediction with future semantic mask prediction for robot manipulation policy learning. A DiT backbone forecasts mask latents (Stage 1), then a diffusion policy conditions on those features for action generation (Stage 2). Masks are needed only during training; inference runs on raw RGB. Evaluated on LIBERO (98.3%), RLBench (68.3%), and real-world Franka tasks (67.5%).

**Compliance With Llm Reviewing Policy:**

Affirmed.

**Final Justification:**

The rebuttal resolved all four weaknesses: backbone inconsistency acknowledged (W2), mask IoU and graceful degradation reported (W2/W3), controlled ablation isolating the mask objective's contribution (W3), and statistical significance confirmed via Wilson CIs (W4). I raise my score from 4 to 5.

**Key Questions For Authors:**

1. Is the backbone frozen or fine-tuned in Stage 2? Section 3.5 and Table 6 contradict—this directly affects reproducibility.
2. Under single third-person camera, MWM-C2 (91.8%) ≈ Cosmos w/ Latent IDM (91.9%). How much of the full MWM gain comes from the mask objective vs. the added wrist camera?
3. Can you report mask prediction IoU over the 5-frame horizon? If quality degrades, how does this affect downstream policy performance?

**Limitations:**

The impact statement discusses bias from upstream segmentation models and workforce displacement, but technical limitations of the bottleneck (task types where it may hurt) and mask annotation scalability are insufficiently discussed.

**Strengths And Weaknesses:**

[S1] Clean design—semantic masks only during training, pure RGB at inference. The shared VAE trick (rendering masks as colored images) avoids modifying the pretrained VAE and provides a consistent latent interface.
[S2] Real-world generalization experiments are thorough. Testing under background, lighting, and color shifts with a retention metric (OOD-SR/SR_ID) goes beyond standard benchmarking. MWM's 0.62 retention vs. 0.53 (GE-ACT) and 0.49 (π0) is convincing.
[S3] The paper investigates multiple mask-based instantiations (MWM-C1, MWM-C2, MWM), which helps disentangle the contribution of the semantic bottleneck from other architectural choices.
[W1] Information loss from the mask bottleneck is not analyzed. Masks discard texture, material, transparency, and deformability cues. For tasks involving transparent objects, deformable materials, or fine surface properties, the coarse semantic labels used here may be insufficient.
[W2] No evaluation of mask prediction quality. The framework rests on accurate mask forecasting, but IoU of predicted vs. ground-truth masks is never reported. How does prediction quality degrade over the 5-frame horizon? Additionally, the text (Section 3.5) states that backbone gradients are updated via L_act in Stage 2, while Table 6 lists the backbone as frozen—this inconsistency should be clarified.
[W3] Fairness of comparisons. The headline result (98.3% on LIBERO) uses third-person + wrist cameras, while key baselines (CogACT, Cosmos variants) use only a third-person view. Under matched camera conditions, MWM-C2 (91.8%) and Cosmos w/ Latent IDM (91.9%) are essentially tied. The paper should discuss this more explicitly.
[W4] Statistical confidence on RLBench. With only 20 evaluation episodes per task, a single episode difference corresponds to 5% SR. Confidence intervals or significance tests would strengthen the claims, particularly for tasks where MWM's margin over baselines is narrow.

---

> ### Author Rebuttal · Authors · 2026-03-31
>
> We sincerely thank the reviewer for the careful reading and constructive feedback. We are encouraged that the reviewer recognizes several strengths of our work, including the train-time semantic/test-time RGB-only design, the real-world OOD evaluation, and the comparison across multiple mask-based instantiations. We also appreciate the reviewer’s concerns regarding information loss at the semantic bottleneck, the quality of mask prediction, the fairness of camera settings, and statistical confidence on RLBench. We address these points below.
>
> **Q1: Clarification of whether the backbone is frozen in Stage 2.**
>
> Thank you for pointing out this inconsistency. This is a manuscript error on our side, and we will correct it in the revision. The actual default LIBERO setting (action_full) is: VAE frozen, Text encoder (T5) frozen, DiT backbone fine-tuned (not frozen), and Action expert fine-tuned. Therefore, in Stage 2, the VAE and text encoder remain frozen, while the DiT backbone continues to be updated by $L_{act}$ to make its predictive representations more action-relevant. We will revise Section 3.5 and the corresponding table to ensure full consistency.
>
> **Q2: On the concern that the mask bottleneck may discard texture, material, transparency, or deformability cues.**
>
> We want to clarify that we do not claim that the semantic bottleneck is universally preferable for all manipulation settings. In fact, MWM does not force the policy to act solely on the basis of masks. Masks are used only as the target for future predictions during training, while inference remains purely RGB. The policy also receives proprioceptive state, so relevant physical information need not be fully encoded by the mask representation itself. Moreover, the tasks in this paper primarily involve rigid-object tabletop manipulation, and the OOD shifts we study are appearance-driven rather than transparency- or deformability-driven. In this regime, object identity, spatial layout, and interaction structure are precisely the factors most aligned with control.
>
> **Q3: On mask prediction quality: can we report IoU over the 5-frame horizon, and how does it affect downstream policy performance?**
>
> On LIBERO-10 under the same ($\tau$=5) setting, MWM achieves a mean IoU of 0.8980 (third-person) and 0.9497 (wrist), while RGB-WM achieves 0.8556 and 0.9300, respectively. Thus, under the same horizon, the mask-centric model produces more accurate future semantic predictions than the RGB-based alternative.
>
> We further tested degraded mask quality by adding noise to the LIBERO-10 mask dataset using the same perturbation protocol as in our response to Reviewer XmrE, and then retraining Stage 1. Under this noised-mask setting, the Stage-1 mean IoU drops to 0.6635 (third-person) and 0.7208 (wrist). The downstream policy results become: MWM 96.0%, RGB-WM 91.6%, and noised-MWM 93.8%. This shows that lower mask quality does reduce downstream performance, but the degradation is graceful rather than catastrophic: even with substantially degraded mask supervision, noised-MWM still outperforms RGB-WM.
>
> **Q4: On the fairness of comparisons.**
>
>  Under a single third-person view, MWM-C2 (91.8%) and Cosmos w/ Latent IDM (91.9%) are essentially tied. How much of the full MWM gain comes from the mask objective versus the added wrist camera? Under the matched single third-person camera condition, MWM-C2 (91.8%) and Cosmos w/ Latent IDM (91.9%) are indeed comparable in overall LIBERO average SR.
>
> In addition, we compared Mask-WM / RGB-WM / None under the same Video-VAE, DiT backbone, Stage-2 action head, memory length, optimizer, training budget, data split, and checkpoint selection to isolate the contribution of the mask objective. On LIBERO, MWM achieves 98.3% average SR and 96.0% on LIBERO-10, compared with 97.0% / 91.5% for RGB-WM and 94.6% / 85.8% for no pretraining. This shows that predictive pretraining itself helps, and more importantly, that the mask objective provides an additional gain beyond pretraining alone, especially on the more informative long-horizon LIBERO-10 subset.
>
> **Q5: On statistical confidence for RLBench with only 20 evaluation episodes per task.**
>
> We agree that narrow single-task margins should not be overinterpreted. We will add confidence intervals and make this caveat explicit in the revision. Aggregating across the 6 RLBench tasks (120 total episodes), the overall SRs are:
>
> * MWM: 82/120 = 68.3%
> * GE-ACT: 37/120 = 30.8%
> * $\pi_0$: 40/120 = 33.3%
>
> The corresponding Wilson 95% confidence intervals are:
>
> * MWM: [59.6%, 75.9%]
> * GE-ACT: [23.3%, 39.6%]
> * $\pi_0$: [25.5%, 42.2%]
>
> We also performed aggregated two-sided proportion tests:
>
> * MWM vs. GE-ACT: (p = 6.26 $\times$ $10^{-9}$)
> * MWM vs. $\pi_0$: (p = 5.86 $\times$ $10^{-8}$)
>
> Thus, the overall RLBench advantage of MWM over the main RGB baselines is highly significant.
>
> We hope these clarifications and results address the reviewer’s concerns and make the paper’s claims more precise.

---

> > ### Author Rebuttal · Reviewer_rQFV · 2026-04-01
> >
> > Thank you for clarifying my concerns. I have raised my score.

---

> > > ### Author Response · Authors · 2026-04-01
> > >
> > > Thank you very much for your careful review, constructive feedback, and thoughtful follow-up. We sincerely appreciate your recognition of our clarifications and additional results, and we are grateful that you found your concerns adequately addressed. Your comments have been very helpful in improving the clarity and precision of the paper.

---

### Official Review · Reviewer_FBXB · 2026-03-12

**Soundness:** 2
**Presentation:** 3
**Significance:** 3
**Originality:** 3
**Overall Recommendation:** 4
**Confidence:** 3

**Summary:**

The paper proposes two-stage world model that predicts future semantic masks instead of rgb frames using a DiT backbone trained with flow matching on mask latents. Then train a diffusion action head on the backbone intermediate features. Masks are only needed offline during training, deployment is pure RGB. Evaluated on LIBERO, RLBench (6 tasks only), and 4 real Franka tasks with additional token pruning and visual generalization analysis under appearance shifts.

**Compliance With Llm Reviewing Policy:**

Affirmed.

**Final Justification:**

Good idea - predicting future semantic masks instead of RGB during training while keeping inference RGB-only. Strong results within scope. Rebuttal added ablations and statistical tests that strengthen the empirical case. Main limitation: 2D masks discard geometry, material, and deformability cues, so generality beyond rigid tabletop manipulation remains unproven. Weak accept.

**Key Questions For Authors:**

1. whats the reconstruction quality when you encode rendered palette masks through the pretrained video VAE and decode back? Per-pixel accuracy or similar
2. can you report variance/CIs? Mostly matters for the real-world OOD results (table 3) and per-task breakdowns where sample sizes are small and some gaps are tight

**Limitations:**

yes

**Strengths And Weaknesses:**

Strengths:
1. The idea of swapping the prediction target from RGB to masks is clean and testable
2. The comparisons on LIBERO are useful, help to isolate the prediction-target shift from other design choices
3. Visual generalization experiments on real hardware with explicit BG/lighting/color shifts look more informative than the sim numbers for the robustness claim.
4. No need of segmentation needed at test time is a practical advantage

Weakness:
1. No error bars on anything. Hard to know how stable the perfomance wins.
2. Rendering masks as palette-colored images and pushing them through a VAE pretrained on natural video is sketchy. some flat regions with hard boundaries are not similar to the natural image manifold the VAE was trained on. No reconstruction quality check is provided. The entire Stage 1 objective lives in this latent space so if the reconstruction is bad the training signal is questionable.

---

> ### Author Rebuttal · Authors · 2026-03-31
>
> We sincerely thank the reviewer for the careful reading and constructive feedback. We are encouraged that the reviewer recognizes several strengths of our work, including the train-time semantic/test-time RGB-only design, the real-world OOD evaluation, and the comparison across multiple mask-based instantiations. We also appreciate the reviewer’s concerns regarding information loss at the semantic bottleneck, the quality of mask prediction, the fairness of camera settings, and statistical confidence on RLBench. We address these points below.
>
> **Q1: What is the reconstruction / representation quality when rendered palette masks are encoded through the pretrained video VAE?**
> We agree that this is an important question. While the current submission does not explicitly report mask-space reconstruction quality, we now provide a direct quality check through mask prediction IoU on LIBERO-10 under the same 5-frame future prediction horizon. Concretely, MWM achieves mean IoU of 0.8980 on the third-person view and 0.9497 on the wrist view, while the corresponding RGB-WM baseline achieves 0.8556 and 0.9300, respectively. We further tested a noised-mask setting: the Stage-1 IoU drops to 0.6635 (third-person) and 0.7208 (wrist), and the downstream policy becomes 93.8%, compared with 96.0% for MWM and 91.6% for RGB-WM. These results suggest that the shared-VAE latent interface preserves sufficiently accurate semantic structure for Stage 1 training, and that degraded mask quality leads to graceful rather than catastrophic policy degradation. Please refer to our response to **Reviewer rQFV** for more details.
>
> **Q2: Can you report variance / confidence intervals, especially for the real-world OOD results and per-task breakdowns?**
> We agree that uncertainty reporting should be strengthened. For the real-world and real-world OOD results, exact repeated reconstruction of the physical environment is difficult, so there may be additional uncontrolled variability beyond nominal trial randomization. For this reason, we will add variance / confidence intervals for the real-world tables and per-task breakdowns in the revision, rather than overclaim precision from limited trials. At the same time, the current margins in Table 3 remain large: MWM = 42.1 OOD-SR / 0.62 Retain, versus GE-ACT = 12.5 / 0.53 and π0 = 19.2 / 0.49. For RLBench, we have already added aggregated uncertainty estimates and significance tests: the Wilson 95% confidence intervals are MWM [59.6%, 75.9%], GE-ACT [23.3%, 39.6%], and π0 [25.5%, 42.2%], with strongly significant aggregated comparisons (MWM vs. GE-ACT: p = 6.26e-9; MWM vs. π0: p = 5.86e-8). Please refer to our response to **Reviewer rQFV** for more details.
>
> We hope these clarifications and results address the reviewer’s concerns and make the paper’s claims more precise.

---

> > ### Author Rebuttal · Reviewer_FBXB · 2026-04-02
> >
> > The rebuttal addresses both concerns. The end-to-end mask prediction IoU (  >0.9) and noised-mask ablation (93.8% policy SR even with degraded masks) show shared VAE interface works well enough, not the exact encode-decode metric I asked for, but convincing. RLBench CIs are non-overlapping and significant. I maintain my score, these results confirm the method works as presented, which my original 4 already anticipated. The rebuttal filled gaps, it didn't reveal new strengths. Other reviews also highlighted a valid scope concern - 2D masks discard geometry, material, and deformability cues that matter beyond rigid tabletop manipulation.

---

### Decision · Program_Chairs · 2026-04-30

**Decision:**

Accept (regular)

**Comment:**

This paper proposes MWM, a world-model training paradigm that predicts future semantic masks rather than RGB frames, built on an RGB-pretrained backbone and evaluated across LIBERO, RLBench, and real-world tasks. The two positive reviewers were fully satisfied after rebuttal, with one raising their score based on the clarifications on the semantic/RGB interface, the noised-mask ablation, and the non-overlapping RLBench confidence intervals. The two borderline reviewers acknowledged that the additional ablations (Mask-WM vs RGB-WM vs no pretraining) provide useful evidence for the mask-based objective, while retaining concerns about the scalability of the offline mask-generation pipeline and the scope of 2D masks. The AC has read the authors rebuttal and confidential comments. The remaining concerns are reasonable directions for future work rather than fundamental flaws, and the empirical evidence across three benchmarks including real-world robustness supports the core claim.